# Quantitative dose-response analysis untangles host bottlenecks to enteric infection

Ian W. Campbell [1,2], Karthik Hullahalli [1,2], Jerrold R. Turner [3,4] & Matthew K. Waldor [1,2,5] ✉

Host bottlenecks prevent many infections before the onset of disease by eliminating invading pathogens. By monitoring the diversity of a barcoded population of the diarrhea causing bacterium *Citrobacter rodentium* during colonization of its natural host, mice, we determine the number of cells that found the infection by establishing a replicative niche. In female mice the size of the pathogen's founding population scales with dose and is controlled by a severe yet slow-acting bottleneck. Reducing stomach acid or changing host genotype modestly relaxes the bottleneck without breaking the fractional relationship between dose and founders. In contrast, disrupting the microbiota causes the founding population to no longer scale with the size of the inoculum and allows the pathogen to infect at almost any dose, indicating that the microbiota creates the dominant bottleneck. Further, in the absence of competition with the microbiota, the diversity of the pathogen population slowly contracts as the population is overtaken by bacteria having lost the critical virulence island, the locus of enterocyte effacement (LEE). Collectively, our findings reveal that the mechanisms of protection by colonization bottlenecks are reflected in and can be generally defined by the impact of dose on the pathogen's founding population.

Diarrhea-causing pathogens remain a global threat to human health, with billions of cases impacting over half of the world's population every year[1]. These pathogens are typically transmitted in food, water, and between infected individuals by the fecal-oral route. However, exposure does not always result in the pathogen establishing a replicative niche in the host, a prerequisite for colonization and thus most disease.

As a natural mouse pathogen, the gram-negative bacterium *Citrobacter rodentium* provides a particularly valuable experimental model to identify the factors governing the earliest events that result in colonization of the mammalian gastrointestinal tract. This enteric pathogen proliferates in the cecum and colon, causes diarrhea, and naturally transmits between mice via coprophagy[2]. *C. rodentium* shares an infection strategy with the human extracellular pathogens

enteropathogenic and enterohemorrhagic *Escherichia coli* (EPEC and EHEC). To colonize the gastrointestinal tract these pathogens depend on a conserved pathogenicity island known as the locus of enterocyte effacement (LEE), which encodes a type three secretion system that mediates pathogen adherence through creation of attaching and effacing (A/E) lesions on host epithelial cells in the colon[3].

While extensive work has focused on the molecular functions of the LEE and the disease caused by A/E pathogens, less is known about the early events that result in establishment of a replicative niche. Dosing studies indicate that ingestion of between a million and a billion bacteria are required for *C. rodentium* to infect mice[4], indicating that one or more barriers likely impede colonization. These barriers collectively create an infection bottleneck by causing

[1]Division of Infectious Diseases, Brigham & Women's Hospital, Boston, USA. [2]Department of Microbiology, Harvard Medical School, Boston, USA. [3]Laboratory of Mucosal Barrier Pathobiology, Department of Pathology, Brigham & Women's Hospital, Boston, USA. [4]Department of Medicine, Harvard Medical School, Boston, USA. [5]Howard Hughes Medical Institute, Chevy Chase, USA. ✉e-mail: mwaldor@research.bwh.harvard.edu

a stochastic contraction the pathogen population. However, the bottlenecks restricting A/E pathogens have not yet been fully characterized because the dramatic expansion of the pathogen population that follows establishment of a replicative niche makes it experimentally difficult to quantify the earliest events required for infection.

Here we inoculated mice with a barcoded, but otherwise isogenic population of *C. rodentium* to differentiate the effects of the initial colonization bottleneck from subsequent population expansion. By monitoring the diversity of barcodes in the population, we determined the number of unique cells from the inoculum (founders) which expand to cause infection (schematized in Fig. 1a). Quantifying the size of the pathogen founding population allowed us to define the bottleneck to colonization and determine when/where the invading pathogen establishes a replicative niche. We discovered that a severe bottleneck restricts colonization and results in a very small number of founders that increase with the size of the inoculum. Multiple factors contribute to the infection bottleneck with the greatest restriction occurring after the pathogen reaches the sites of infection. Notably, the magnitude of the infection bottleneck is largely explained by a combination of stomach acid and microbiota, where the latter is approximately a million times more protective.

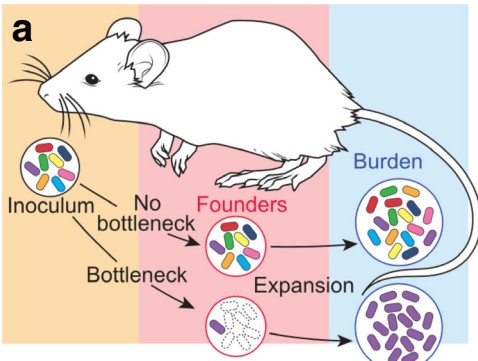

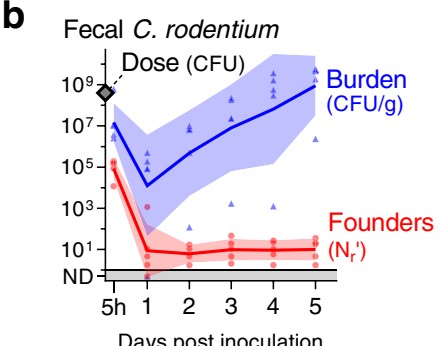

**Fig. 1 | A constrictive bottleneck impedes *C. rodentium* enteric colonization.**
**a** Experimental strategy. To quantify the contraction (bottleneck) of the *C. rodentium* population during colonization a barcoded library of bacteria was orally administered to mice. The number of unique cells from the inoculum (founders) that expand to produce the observable population (burden) was determined from the frequency and diversity of barcodes. **b** 5 cohoused, adult, female, C57BL/6 J mice were administered $4 \times 10^8$ CFU STAMP-CR253 by oral gavage and the *C. rodentium* population was monitored in the feces for 5 days post inoculation (p. i.). Dose and burden were enumerated by serial dilution and plating; founders were determined by STAMP. Lines and error are geometric means and standard deviation. Bacteria not detected (ND) counted as 0.5, CFU colony forming units, g grams. Validation of STAMP libraries in Supplemental Fig. 1. Source data are provided as a Source Data file.

## Results

### A small number of *C. rodentium* founders initiates enteric infection

To enable monitoring of the pathogen population's diversity during infection, we introduced short, random, -20 nucleotide DNA tags (barcodes) at a neutral location in the *C. rodentium* genome. As previously described[5], monitoring barcode diversity using high-throughput DNA sequencing and the STAMP (Sequence Tag-based Analysis of Microbial Populations) computational framework can quantify the constriction of the pathogen population that often occurs during establishment of infection (schematized in Fig. 1a). We created two independent STAMP libraries of barcoded bacteria. Library "STAMP-CR253" contains 253 unique barcodes integrated in the intergenic region between genes *ROD_05521* and *selU*. The neutrality of the barcode insertions was confirmed by measuring growth in lysogeny broth (LB; Supplemental Fig. 1a). Library "STAMP-CR69K" contains approximately 69,000 unique barcodes inserted into the genome on a Tn7 vector, which integrates at a neutral site downstream of the *glmS* gene[6,7]. While the libraries were not directly compared, both yielded similar results in our studies.

To validate that these barcoded STAMP libraries can quantify the population effects of a bottleneck, we created in vitro bottlenecks by plating serial dilutions of the libraries grown in culture. The number of colony forming units (CFU) per plate provides a true measure of the number of founders, i.e., the number of cells from the initial population (culture) that gave rise to the observed population (plated colonies). Bacteria were harvested from the plates, the barcodes were amplified and sequenced, and barcode frequencies were analyzed using the recently updated STAMP analysis pipeline "STAMPR"[8]. The size of the founding population (founders) was calculated by comparing the diversity and frequency of barcodes recovered from plated samples to those in the initial cultures. There was a strong correlation between the counted founders (CFU) and the calculated founders ($N_r$ for STAMP-CR253 and $N_s$ for STAMP-CR69K) up to $10^4$ founders (Supplemental Fig. 1b). These data were also used as standard curves to increase the resolution of the experiments described below to approximately $10^6$ founders.

Contraction of a barcoded population during colonization changes the frequency and number of barcodes relative to the inoculum. To determine when a *C. rodentium* infection is founded, C57BL/6 J (B6) mice were orally gavaged with $4 \times 10^8$ CFUs (enumerated by serial dilution and plating). Remarkably, despite this relatively large dose, within 24 hours (h) there was an average of only 9 founders (geometric mean), and as few as one founder per mouse (a single barcode; Fig. 1b). Thus, only -1 of every $4 \times 10^7$ cells in the inoculum establishes infection, revealing that host bottlenecks result in a massive constriction of the pathogen population. Beyond 24 h, the founding population remained stable at -10 founders. The diminutive *C. rodentium* founding population indicates that the vast majority of the inoculum does not survive to give rise to detectable offspring and is thus either killed by the host or passes through the intestine and is excreted in feces. Consistent with the latter possibility, 5 h after inoculation there were $1 \times 10^7$ CFU *C. rodentium* per gram of feces with $9 \times 10^4$ founders, suggesting that at this early point a numerous and diverse population has already reached the colon and cecum but failed to become founders. Surprisingly, the contraction of the pathogen population continued beyond 5 h, when the pathogen had already reached the principal sites of colonization. Despite the profound bottleneck to infection, the -10 founders were capable of replication, and by 5 days post inoculation the *C. rodentium* burden in the feces was on average $9 \times 10^8$ CFU/gram. Together, these observations reveal that there is a severe bottleneck to infection with this natural, mouse enteric pathogen; however, even though the restrictive bottleneck leaves a founding population that is a miniscule fraction of the inoculum, the founders robustly replicate, creating a total pathogen burden that ultimately exceeds the inoculum (Fig. 1b).

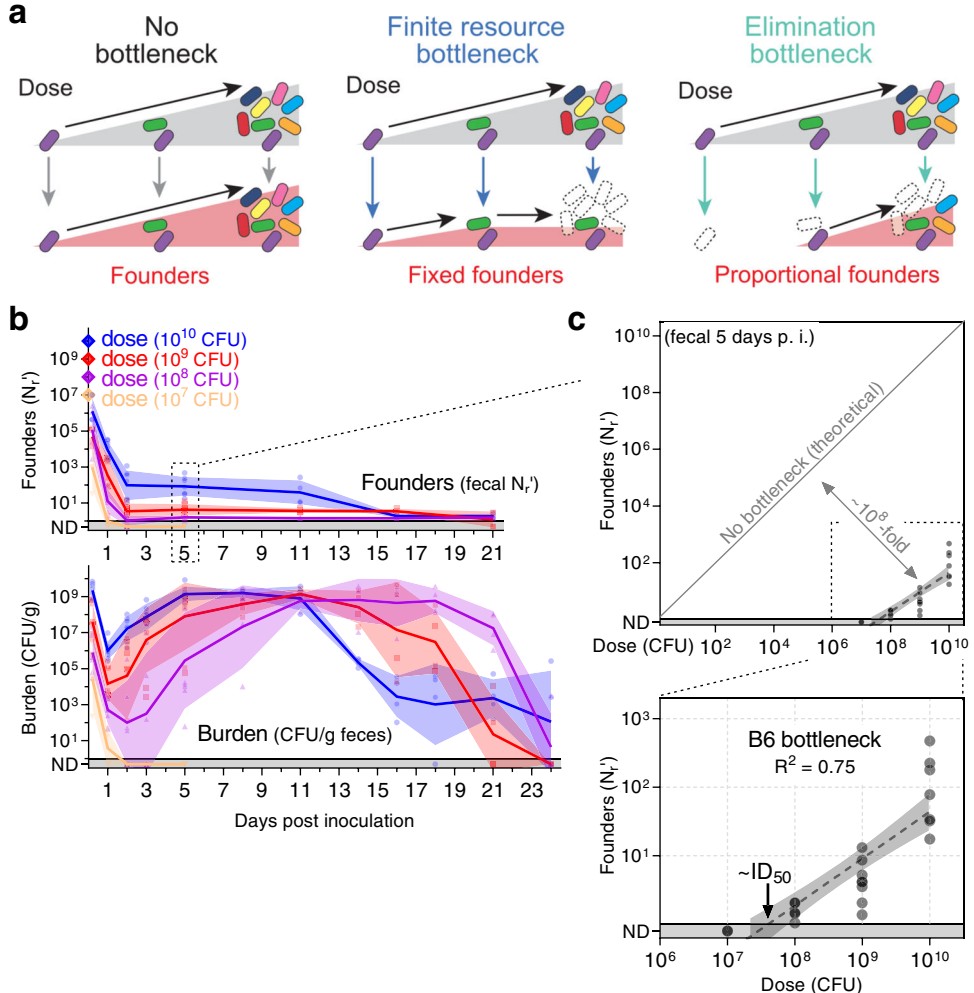

**Fig. 2 | The *C. rodentium* bottleneck is defined by a fractional relationship between dose and founders. a** Models for the relationship between dose and founding population. In the absence of a bottleneck, all bacteria from the inoculum become founders. If the inoculum contracts due to the limited availability of finite nutrients or niches (e.g. iron, sugar, binding sites), the diversity of the population and thus the size of the founding population will remain fixed once those nutrients are saturated. If increasing dose increases the number of founders, then the underlying mechanism is not due to a limited resource; instead, the bottleneck acts proportionally on the inoculum by eliminating potential founders. **b**, **c** C57BL/6 J mice were inoculated with doses ranging from $10^7$ to $10^{10}$ CFU of STAMP-CR253 and the *C. rodentium* population was monitored in the feces (geometric means and standard deviations; ND not detected counted as 0.5). Additional shedding analysis in Supplemental Fig. 2. **c** The bottleneck impeding B6 colonization is described 5 days post inoculation by comparing dose and founders with a linear regression of the $\log_{10}$-transformed data (regression line with 95% confidence intervals; not detected counted as 0.8; *x*-intercept "ID$_{50}$" $10^{7.2}$–$10^{7.9}$ CFU). 4–8 animals per dose. Source data are provided as a Source Data file.

## The size of the founding population increases with dose

We reasoned that determining how dose impacts the number of founders could provide insight into the mechanisms underlying the bottleneck[9,10]. For example, one explanation proposed for the *C. rodentium* bottleneck is that it is created by finite niches or resources (e.g., sugar or amino acids) whose scarcity limits the size of the population[11,12]. At doses where the pathogen saturates this limited resource, the 'finite resource' hypothesis predicts that increasing dose will not increase the number of founders (schematized in Fig. 2a). An alternate possibility is that the bottleneck eliminates potential founders through a mechanism such as acid killing in the stomach[13,14], which is expected to result in a founding population that increases with dose.

To characterize the bottleneck, we orally inoculated B6 mice with *C. rodentium* doses ranging 1000-fold from $10^7$ to $10^{10}$ CFUs. Doses ≥$10^8$ CFUs led to infection, with the founding population decreasing for ~2 days before reaching a steady value that persisted until the infection began clearing, as indicated by a simultaneous decrease in the total population (burden) and founding population (Fig. 2b). Lower doses of *C. rodentium* resulted in fewer founders and a longer period to

reach peak shedding, with a correspondingly longer time from inoculation to pathogen elimination. As the delay in shedding at lower doses correlated with the delay in clearance, all mice were infected for a similar number of days and had similar total fecal burdens, regardless of dose (Supplemental Fig. 2).

The founding population was small in number. Even at the maximum inoculum of $10^{10}$ CFUs relatively few founders were detected (83, geometric mean; Fig. 2b). While founders were never numerous, increasing the size of the inoculum always increased the size of the founding population. The bottleneck eliminated a proportion of the *C. rodentium* population, resulting in a founding population that scaled with dose (increasing dose 100-fold also increased founders ~100-fold). These observations indicate that the number of founders is likely not dictated by limited space or resources, contradicting the finite resource hypothesis (Fig. 2a).

As an increase in dose resulted in a proportional increase in the number of founders, we represented their relationship as a line by plotting $\log_{10}$-transformed dose and founding population data from 5 days post inoculation (Fig. 2c). This line indicates that the bottleneck

is not fixed, but rather functions by eliminating a fraction of potential founders, as schematized in Fig. 2a ('elimination bottleneck'). Since our findings conform to a simple fractional relationship between dose and founding population, we will use this relationship to define the bottleneck: in B6 mice 1 of every ~$10^8$ inoculated *C. rodentium* establish a replicative niche.

The *x*-intercept of the log-linear relationship between dose and founders can be used to calculate the dose at which we expect 1 founder. This dose corresponds to the $ID_{50}$ - the dose that leads to infection of ~50% of animals. Thus, for *C. rodentium* infection, the $ID_{50}$, a critical parameter describing a pathogen's infectivity, is a property biologically defined by the infection bottleneck. For B6 mice, the *x*-intercept of this line is between $10^{7.2}$ and $10^{7.9}$ CFU (95% confidence-intervals) and explains why infection did not result from an inoculum of $10^7$ CFU (Fig. 2b). Surprisingly, even though *C. rodentium* is a natural mouse pathogen, at least ~100-million organisms are required to routinely establish infection.

### Stomach acid contributes a 10- to 100-fold bottleneck to *C. rodentium* colonization

We next probed the contribution of stomach acid to the highly restrictive B6 enteric colonization bottleneck. The acidity of the stomach is thought to be a potent barrier against ingested bacteria; human studies find that taking stomach acid reducing drugs increases the risk of contracting multiple enteric pathogens[15]. Notably, it has been observed that eliminating stomach acid decreases the minimum infectious dose for *C. rodentium* and increases the size of the founding population[13,14]. Further, acid is mechanistically consistent with the fractional relationship which we observe between dose and founding population (Fig. 2). To test the role of stomach acid in restricting *C. rodentium* enteric colonization, we treated mice with the fast-acting, irreversible H2-antagonist Loxtidine (aka Lavoltidine)[16]. 3–5 h after Loxtidine treatment, the pH of the stomach rose from 2.5 to 4.7 (Fig. 3a). Importantly, a pH of 2.5 sterilized $10^{10}$ CFUs of *C. rodentium* in under 15 minutes (min), whereas pH 4.7 did not kill *C. rodentium* even after a 1 h exposure (Fig. 3b).

Loxtidine treatment prior to inoculating B6 mice resulted in infection at a lower dose, a higher pathogen burden in the feces 1 day post inoculation, and more founders on day 5 (Fig. 3c, d). The fractional relationship between dose and founding population was also observed in the absence of stomach acid, but the line depicting this relationship was shifted upward. Loxtidine treatment increased the number of *C. rodentium* founders approximately 10-fold at every dose, reducing the $ID_{50}$ computed from the founding population from $10^{7.3}$ to $10^{5.4}$ CFUs. Thus, stomach acid significantly contributes to the bottleneck restricting *C. rodentium* colonization. However, the magnitude of stomach acid's contribution is relatively small, between 10- and 100-fold of the observed ~$10^8$-fold B6 bottleneck to *C. rodentium* colonization. In the absence of stomach acid, the *C. rodentium* population constricts >$10^6$-fold prior to establishing a replicative niche, indicating that other factors must more potently contribute to the bottleneck.

### Constriction of the *C. rodentium* inoculum occurs distal to the stomach, at the sites of infection

To further define the *C. rodentium* population dynamics and host barriers that accompany establishment of infection, we probed where and when the bottleneck occurs. Five days post-inoculation, the largest pathogen burdens were detected in the cecum and distal colon, with less numerous populations in the small intestine (SI) (Fig. 4a), consistent with previous observations[17]. Within individual mice (intra-mouse) the cecum, colon, and feces contained related populations of *C. rodentium*, with approximately the same number of founders and similar barcodes (Fig. 4b–e). Importantly, the near identity of the barcodes found in the fecal population to those in the cecum and colon indicates that fecal samples can be used to report

on the pathogen population at these primary infection sites, facilitating longitudinal monitoring. While intra-mouse populations were related, comparisons of barcodes between cohoused mice (inter-mouse) inoculated with the same inoculum revealed that each mouse contained a distinct *C. rodentium* population (Fig. 4c–e). The distinct identities of the founding populations in each of five cohoused, co-inoculated mice was apparent when comparing pathogen barcode frequencies with principal component analysis (PCA), where intra-mouse samples formed their own tight clusters (Fig. 4c). Similarly, analysis of barcode genetic distances showed that the intra-mouse pathogen populations were highly similar (low genetic distance), whereas they were dissimilar to the populations in cohoused, co-inoculated mice (Fig. 4d, e). A notable exception were the *C. rodentium* populations from some SI samples that were more closely related to cage-mates than other intra-mouse samples, likely reflecting recent inter-mouse exchange via coprophagy (Supplemental Fig. 3). These data suggest that despite the consumption of *C. rodentium*-laden feces, *C. rodentium* infection leads to super-colonization resistance at the primary infection sites in the cecum and colon, preventing transmission to cohoused, co-infected mice.

To test this super-colonization resistance hypothesis, we separately infected two groups of 'seed' mice with different sets (A and B) of barcoded *C. rodentium* (Supplemental Fig. 4a). At the peak of colonization in the seed mice, 7 days post-inoculation, they were cohoused for 16 h along with an uninfected 'contact' mouse, three mice per cage. After 16 h, the mice were separated back into 3 cages containing mice originally inoculated with the A barcodes, inoculated with the B barcodes, or uninoculated. No transmission of barcodes was detected between the animals originally inoculated with the A and B barcodes (Supplemental Fig. 4b), confirming that *C. rodentium* infection prevents super-colonization. In marked contrast, the contact mice became infected with founders from seed A and/or B, demonstrating the ready transmission of *C. rodentium* from infected to uninfected mice. Furthermore, the co-infection of contact mice with barcodes from A and B confirms a previous report from super-infection experiments in mice lacking a microbiota[18] that immunity to super-colonization takes time, providing a window for co-colonization. Importantly, super-colonization resistance indicates that founders are more likely to originate from the inoculum than other cohoused, infected animals.

Based on the high burdens of *C. rodentium* in the cecum but not the colon during the first 3 days following inoculation, prior studies proposed that infection begins with pathogen expansion in the cecum, followed by subsequent spread to the colon[17]; a hypothesis that is consistent with the closely related intra-mouse *C. rodentium* populations that we observe in the cecum and colon 5 days after inoculation (Fig. 4a–e). To determine when and where *C. rodentium* initiates infection, we monitored the luminal and adherent *C. rodentium* populations in the cecum and colon. Within the first 5 h a large burden (>$10^7$ CFU) and numerous founders (>$10^5$ $N_r'$) were detected in both locations (Fig. 4f, g). Since a large founding population was observed in the cecum and colon early after inoculation, we can discount the model that the primary bottleneck occurs proximal to these locations (e.g., stomach acid or bile). The number of founders and the total burden contracted over the first 24 h, resulting in small (<$10^5$ CFU) dissimilar (genetic distance >0.4) populations in the cecum and colon one day post inoculation. Expansion was detected first in the cecum, on day 2. Concomitant with cecal expansion, the populations in the cecum and colon became increasingly similar; i.e., the genetic distance between the populations became smaller. The most plausible model to fit these data (depicted in Fig. 4h) is that (1) within hours many bacteria pass through the stomach, reaching the cecum and colon, and then (2) these populations diverge as they separately constrict, and finally, (3) spread to both locations when a small number of founders begin to replicate. We propose that the initial population

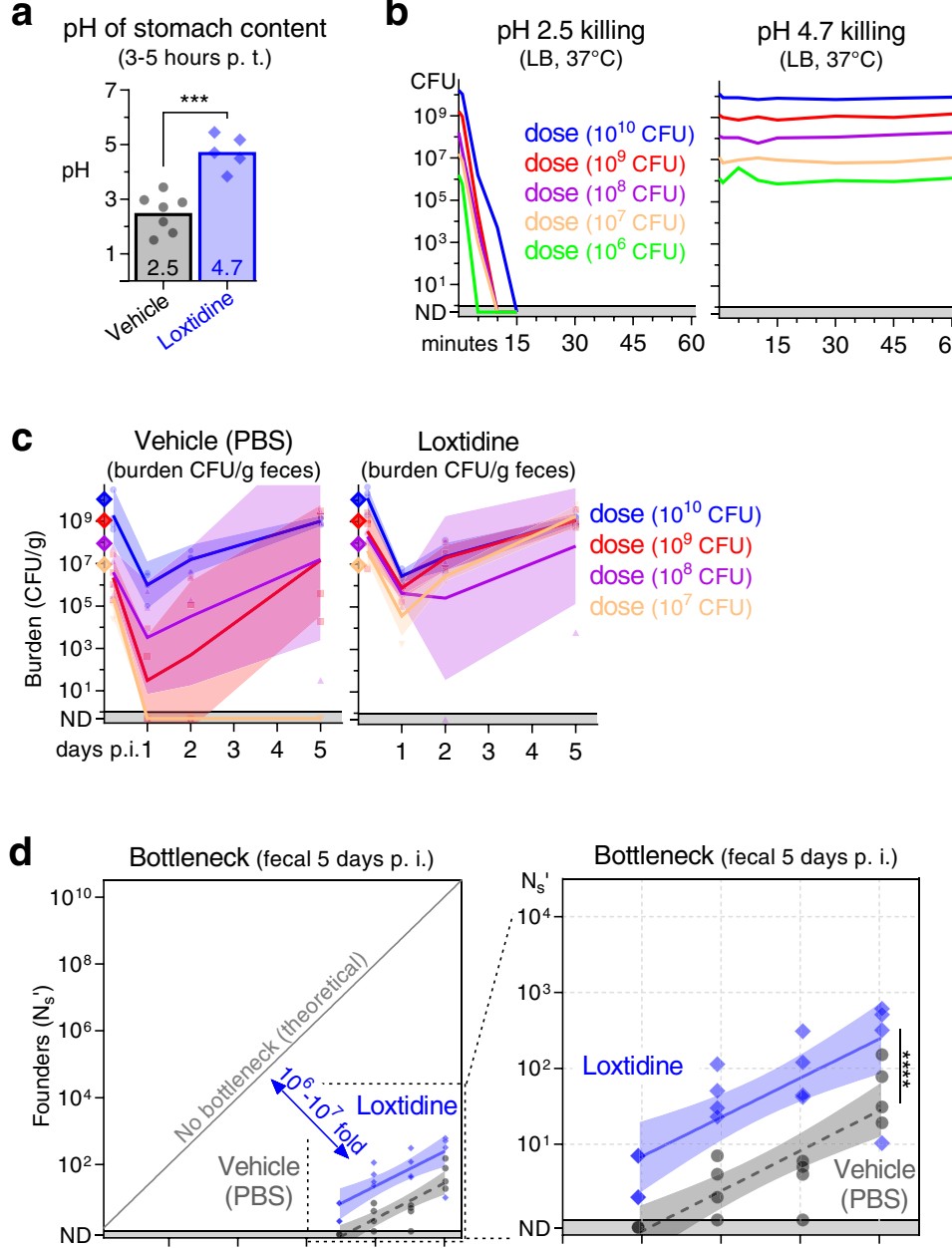

**Fig. 3 | Stomach acid constricts the *C. rodentium* population by 10- to 100-fold.**
**a** The effect of Loxtidine on stomach acid in C57BL/6 J mice 3–5 h after intraperitoneal administration of 1 mg in 0.1 ml PBS. pH determined post-mortem in aspirated stomach fluid. Boxes arithmetic mean (2.5 for mock and 4.7 for Loxtidine). Two-tailed *t* test with *p*-value 0.0002. Animals are 7 (vehicle) and 5 (Loxtidine). **b** The acid tolerance of STAMP-CR69K in culture measured by diluting cells in LB at pH 2.5 or 4.7 and incubating at 37 °C with shaking. pH 2.5 sterilized $10^{10}$ CFU in 15 min. **c**, **d** 3–5 h after intraperitoneal administration of PBS (vehicle) or Loxtidine (1 mg), C57BL/6 J mice were orally gavaged with doses ranging from $10^7$ to $10^{10}$ CFU of STAMP-CR69K. **c** Bacterial burden monitored in the feces for 5 days following inoculation (geometric means and standard deviations; ND not detected counted as 0.5). **d** Bottleneck to colonization measured 5 days post inoculation by comparing dose and founders with linear regression of the $\log_{10}$-transformed data (regression line and 95% confidence intervals; significance compares elevation with *p*-value $5.5 \times 10^{-7}$; not detected counted as 0.8). 4 animals per dose per group. Source data are provided as a Source Data file.

expansion begins in the cecum and then spreads to the colon, but we cannot rule out the opposite directionality because we were unable to serially sample the internal populations from a single mouse. However, displacement of the cecal population by bacteria from the colon seems unlikely because it would require non-flagellated *C. rodentium* to move against the bulk flow of the gut and thus we favor the model that infection initiates in the cecum.

## C3H/HeOuJ mice have a less restrictive bottleneck than C57BL/6 J

We next interrogated the host's contribution to the bottleneck impeding *C. rodentium* colonization by quantifying the bottleneck in a more disease susceptible genotype of mice. While *C. rodentium* causes self-limited diarrhea in B6 mice, infection leads to a lethal diarrheal disease in C3H/HeOuJ (C3Ou) mice (Fig. 5a, b)[19]. We found

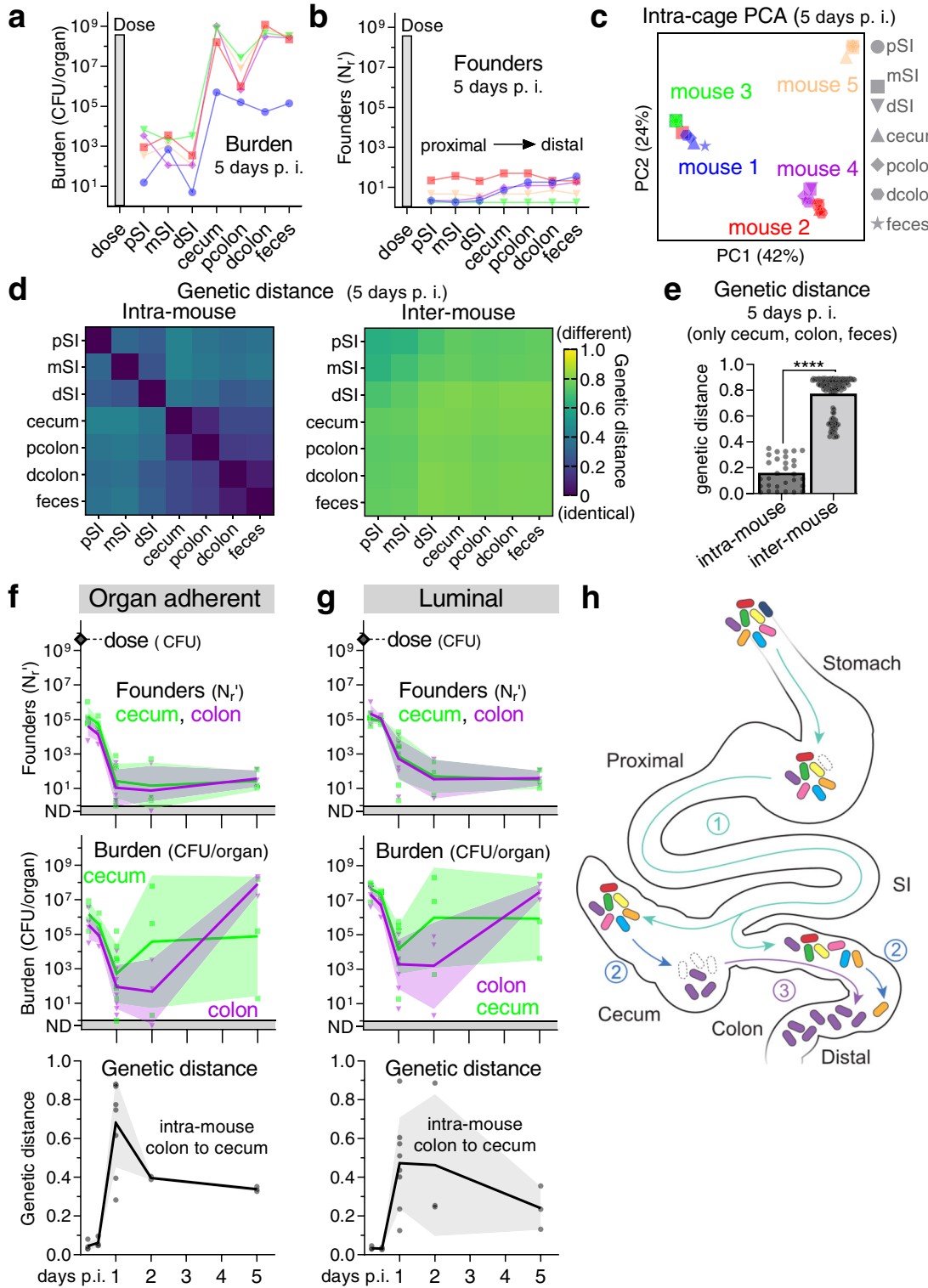

that increased vulnerability to disease correlated with a less restrictive bottleneck. *C. rodentium* is 10- to 100-fold more infectious in C3Ou than B6 mice, infecting at a ~10-fold lower dose and producing ~10-times more founders at every dose (Fig. 5c). While the bottleneck was relaxed in C3Ou mice, a fractional relationship remained between dose and founding population, suggesting a similar underlying mechanism restricts colonization in both mouse genotypes. Also, as in B6 animals, higher doses and more founders accelerated the dynamics of pathogen shedding in C3Ou mice (Fig. 5a). These observations demonstrate that in addition to dose,

the size of the founding population is determined in part by host genetics, which may impact the bottleneck through several mechanisms. Notably, changing host genotype caused a more lethal disease while only alleviating ~10-fold of the ~10[8]-fold B6 bottleneck.

### The bottleneck to *C. rodentium* enteric colonization is microbiota dependent
As shown above, a large portion of the restrictive, fractional, B6 bottleneck to *C. rodentium* colonization occurs distal to the stomach, at the chief sites of infection in the cecum and colon. These data strongly

**Fig. 4 | Infection is initiated by related populations of *C. rodentium* in the cecum and colon. a–e** *C. rodentium* populations in whole organ homogenates from 5 cohoused (intra-cage) C57BL/6J mice 5 days post inoculation with $4 \times 10^8$ CFU of STAMP-CR253. Within a mouse (intra-mouse) the *C. rodentium* populations at the primary sites of colonization (cecum, proximal colon, distal colon, feces) share founders (number, identity, and frequency of barcodes). **a, b** Lines connect intra-mouse samples. **c** Clustering of barcode populations by principal component analysis (PCA). **d, e** Relatedness determined by comparing the barcode frequencies by genetic distance (arithmetic means) with zero indicating no difference between populations (identical). **e** Two-tailed *t* test with *p*-value $5.8 \times 10^{-48}$. p proximal, m mid, d distal, SI small intestine. Heatmap depicting genetic distance relationships of all intra-cage populations in Supplemental Fig. 3. Source data are provided as a Source Data file. **f, g** To determine when/where *C. rodentium* establishes a replicative niche, C57BL/6J mice were orally gavaged with between $3 \times 10^9$ and $6 \times 10^9$

CFU STAMP-CR253. Following dissection, the cecum and colon were flushed to separate organ adherent (**f**) and luminal (**g**) bacteria. Burden and founders display geometric means and standard deviations. Bacteria not detected (ND) counted as 0.5. Relatedness of populations was determined by comparing the barcode frequencies of colon and cecal populations from within the same animal (intra-mouse) by genetic distance (arithmetic mean and standard deviation). 22 animals. Source data are provided as a Source Data file. **h** Model depicting how related *C. rodentium* populations could initiate infection in both the cecum and colon: (1) the inoculum minorly constricts passing through the stomach and SI to deposit diverse populations in the cecum and colon, (2) the populations in the cecum and colon contract separately over the first 24–48 h becoming dissimilar, and then (3) expansion occurs in either the cecum or colon moving to both locations. We depict the movement from cecum to colon as we judge this to be more likely, but the opposite is possible.

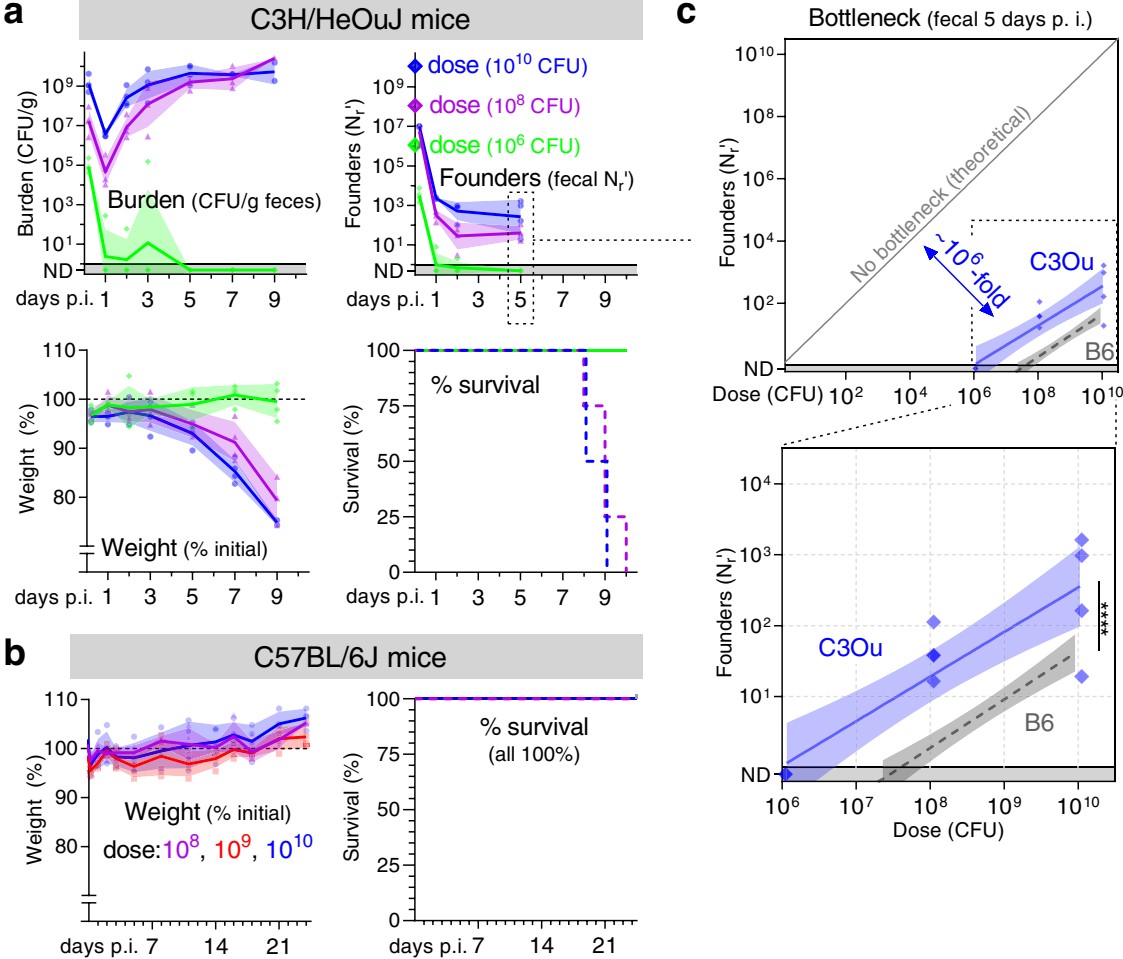

**Fig. 5 | Host genotype impacts the bottleneck to *C. rodentium* colonization.** C3H/HeOuJ (**a**) or C57BL/6J (**b**) mice were inoculated with doses ranging from $10^6$ to $10^{10}$ CFU of STAMP-CR253. The *C. rodentium* population was monitored in the feces (geometric means and standard deviations; ND not detected counted as 0.5) and animal health assessed by weight loss (percent compared to pre-inoculation; arithmetic means and standard deviations) and body condition. For survival, lines are percent of initial animals not moribund. **c** The bottleneck to C3Ou and B6 colonization is described 5 days post inoculation by comparing dose and founders with a linear regression of the $\log_{10}$-transformed data (regression line with 95% confidence intervals; significance compares elevation with *p*-value $5.5 \times 10^{-7}$; not detected counted as 0.8). B6 bottleneck data is repeated from Fig. 2. 4 animals per dose. Source data are provided as a Source Data file.

suggest that the principal step limiting colonization occurs during the pathogen's establishment of a replicative niche in the cecum and/or colon. One factor present at these sites and previously linked to limiting *C. rodentium* colonization is the microbiota[12,18]. We therefore tested whether acute microbiota depletion eliminated the bottleneck to *C. rodentium* colonization. Treating mice with the antibiotic streptomycin for the 3 days prior to inoculation with streptomycin-resistant

*C. rodentium* greatly accelerated pathogen population expansion, with mice shedding $>10^9$ CFUs per gram of feces within the first day (Fig. 6a). Further, streptomycin pretreatment almost completely ablated the bottleneck, with colonization at doses as low as ~100 CFUs; at this low dose, we measured an average of 25 founders 5 days post inoculation, indicating that *C. rodentium* experiences less than a 10-fold bottleneck following microbiota depletion (Fig. 6b). Significantly,

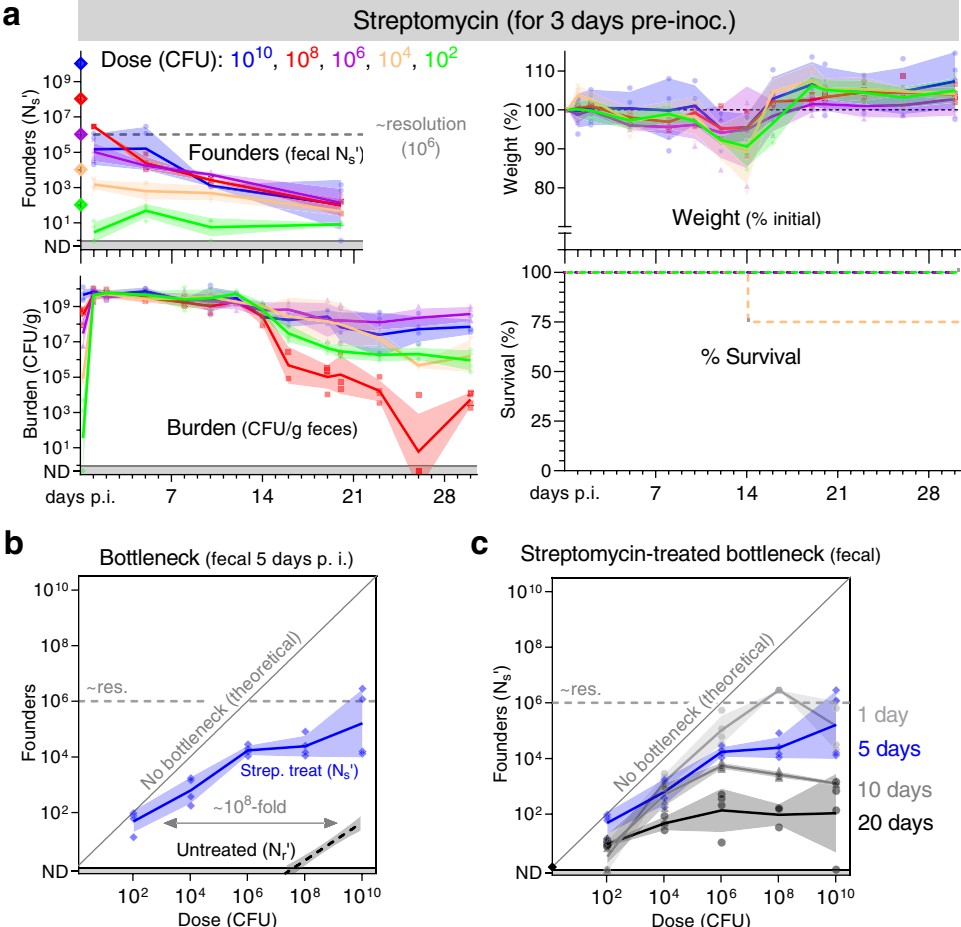

**Fig. 6 | Streptomycin treatment ablates most of the bottleneck preventing *C. rodentium* colonization.** The microbiota of conventional C57BL/6 J mice was reduced by treatment with the antibiotic streptomycin for the 3 days prior to inoculation with a streptomycin resistant library of STAMP-CR69K. **a** Founding population and bacterial burden monitored in the feces, and (**b, c**) the bottleneck to colonization measured by comparing dose and founders (geometric means and standard deviations; resolution limit is ~$10^6$ founders). (**a**) Animal health monitored by weight loss (percent compared to pre-inoculation; arithmetic means and standard deviations) and body condition. For survival, lines are percent of initial animals not moribund. Untreated B6 bottleneck data is repeated from Fig. 2 for comparison. 4 animals per dose. Streptomycin treatment does not impact stomach acidity (Supplemental Fig. 5). Source data are provided as a Source Data file.

streptomycin treatment does not alter the acidity of the animal's stomach (Supplemental Fig. 5). Since an ~10-fold bottleneck remains after microbiota depletion and an ~10-fold bottleneck is stomach acid dependent (Fig. 3d), these data suggest that the combination of the microbiota and stomach acid can account for the majority of factors restricting *C. rodentium* colonization.

To confirm that streptomycin's ablation of the bottleneck to *C. rodentium* colonization occurs because of microbiota depletion rather than an off-target effect, we also determined the bottleneck in B6 mice lacking a microbiota (germ-free). In germ-free mice, like streptomycin pretreated animals, there was almost no bottleneck to *C. rodentium* colonization (Fig. 7a, b). Animals lacking a microbiota were colonized at a dose of 150 CFU and shed numerous *C. rodentium* within 1 day of inoculation. Together, experiments with germ-free and streptomycin-pretreated mice reveal that the primary barrier to enteric colonization is linked to the microbiota.

Microbiota disruption also impaired the capacity of mice to clear *C. rodentium* infection (Figs. 6a, 7a)[12]. Pathogen burden in the feces of germ-free mice did not decrease over time, in marked contrast to mice with an intact microbiota (specific pathogen free; SPF). Similarly, most cages of streptomycin-pretreated mice failed to clear the pathogen, with heterogeneity presumably caused by variation in the rebound of the microbiota after streptomycin treatment (Fig. 6a). Despite high fecal burdens, germ-free animals only exhibited mild diarrhea and did

not lose weight for the 30 days of observation. These data indicate that the microbiota is the primary impediment to *C. rodentium* replication in the gastrointestinal tract, antagonizing the pathogen's capacity to initiate a replicative niche and promoting its clearance.

In germ-free and streptomycin pretreated animals the number of *C. rodentium* founders ceased to be fractionally related to dose; doses ranging 10,000-fold, from $10^6$ to $10^{10}$ CFUs, all yielded a similar number of founders 5-days post-inoculation (Figs. 6b, 7b). These data suggest that there is an upper limit to the size of the *C. rodentium* founding population of ~$10^5$ on day 5 (i.e., a bottleneck caused by limited resources as illustrated in Fig. 2a). Furthermore, in the absence of a microbiota dependent bottleneck, the maximum size of the founding population continuously decreased for the 20 days of observation (Figs. 6c, 7c). Although there was no contraction in the *C. rodentium* burden following infection of germ-free animals, the maximum number of founders decreased from ~$10^5$ on day 5 to ~$10^2$ on day 20 (Fig. 7a, c). A decrease in diversity without a decrease in abundance suggests that *C. rodentium* adapts to the germ-free environment, introducing a new bottleneck caused by intra-pathogen competition.

To test the hypothesis that *C. rodentium* evolved during colonization of germ-free animals, we sequenced the genomes of single *C. rodentium* colonies isolated from infected SPF or germ-free mice 5 or 20 days post inoculation. 5 days post-inoculation, the *C. rodentium*

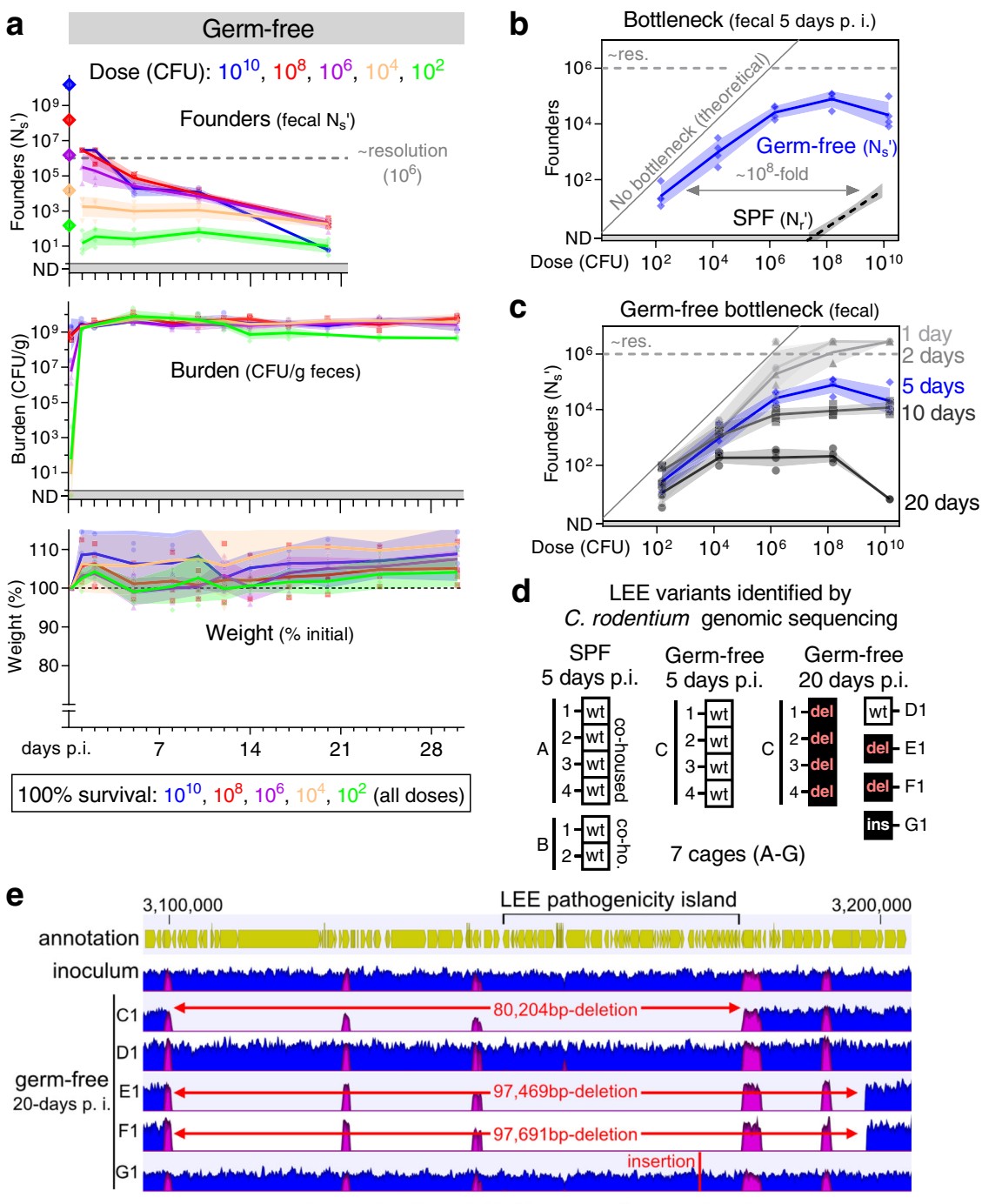

**Fig. 7 | The bottleneck to *C. rodentium* colonization is microbiota dependent.**
Germ-free C57BL/6 J mice were orally inoculated with doses ranging from $10^2$ to $10^{10}$ CFU of STAMP-CR69K. 4 animals per dose, cohoused with animals receiving the same dose in sterile cages. Measurement of germ-free stomach acidity in Supplemental Fig. 5. Source data are provided as a Source Data file. **a** Bacterial burden and founding population monitored in the feces (geometric means and standard deviations; resolution limit is ~$10^6$ founders). Animal health monitored by weight loss (percent compared to pre-inoculation; arithmetic means and standard deviations) and body condition. No animals became moribund. **b**, **c** Bottleneck to colonization measured by comparing dose and founders (geometric means and standard deviations). SPF B6 bottleneck data is repeated from Fig. 2 for comparison. **d** To determine if colonization was accompanied by changes in the *C.*

*rodentium* genome, whole genome sequencing was performed on 3 clones (colonies) from the STAMP-CR69K input library and compared to clones isolated from feces of infected mice (1 colony per mouse). Boxes represent the genome status of the LEE pathogenicity island. No LEE genomic changes wild-type (wt), deletion of the entire LEE region (del), insertion within the LEE (ins). Mice with a conventional microbiota SPF specific pathogen free. Other genomic changes listed in Supplemental table 1. **e** Depiction of a ~100,000 base-pair region of the *C. rodentium* genome containing the LEE pathogenicity island. Read depth from STAMP-CR69K inoculum and 5 clones isolated after 20 days passage in otherwise germ-free animals. Large deletions in 3 of 5 clones revealed by lack of specifically mapped reads in regions of up to 97,691 base-pairs. Non-specific reads map to multiple loci in the genome (primarily transposons).

genomes isolated from SPF and germ-free mice were similar to the inoculum, with 6/10 colonies lacking detectable variations (Supplemental table 1). These data indicate that the initial contraction of the *C. rodentium* population observed during establishment of infection is not caused by selection of a genetically distinct subpopulation of the inoculum. By contrast, 20 days growth in the absence of a microbiota was always accompanied by changes in the *C. rodentium* genome. Notably, *C. rodentium* with structural variations in the LEE pathogenicity island became dominant in 4 out of 5 cages of infected germ-free animals (Fig. 7d, Supplemental table 1). These variations included large deletions of up to 97,691 bps (Fig. 7e, isolate from mouse F1) that eliminated the entire island, which is essential for colonization of SPF mice[20]. These genome alterations suggest that in the absence of a microbiota, a common mechanism for *C. rodentium* adaption to the host environment is to lose the LEE pathogenicity island. Thus, we conclude that competition among *C. rodentium* constricts the diversity of the population in the absence of a microbiota-dependent bottleneck, with organisms that lose the LEE virulence island outcompeting bacteria possessing the LEE. Additional mutations were also detected in the *C. rodentium* isolated on day 20 from germ-free animals, including in the galactonate operon, which have previously been observed in *Escherichia coli* colonizing microbiota depleted mice[21] (Supplemental Table 1). Thus, there may be common evolutionary strategies for pathogenic and non-pathogenic bacteria to adapt to growth without competition in the host intestine.

Collectively these experiments show that of the multiple host factors protecting against enteric infection, the microbiota is by far the most restrictive. Diminution of the microbiota markedly increases host susceptibility, permitting infection at almost any dose. In the absence of competition with the microbiota, a new slow-acting bottleneck constricted the *C. rodentium* population as the pathogen evolved increased fitness, notably through loss of the LEE pathogenicity island.

## Discussion

Here, we investigated the host bottlenecks that protect against *C. rodentium* colonization. We found that the pathogen population undergoes a severe and slow-acting bottleneck. The size of the founding population scaled proportionally with dose: increasing the inoculum 100-fold increased the number of founders ~100-fold. Since increasing pathogen dose also increased the number of founders, we conclude that the bottleneck to colonization is not caused by the limited availability of finite resources or niches. We used the relationship between dose and founding population to interrogate the factors that create the bottleneck to *C. rodentium* colonization and discovered that the bottleneck occurs in at least two distinct sites: an initial stomach acid barrier kills 90–99% of the inoculum and the microbiota removes an additional 99.99999% after the pathogen reaches the cecum. Based on these results we propose a microbiota dependent factor creates the primary bottleneck by inhibiting *C. rodentium's* infectivity. Overall, our findings suggest that quantifying the impact of dose on founding population is a powerful framework for understanding the protection provided by infection bottlenecks.

While *C. rodentium* is capable of remarkable expansion after establishing a replicative niche, this pathogen has unexpectedly low infectivity in experimental conditions and requires a high dose to colonize. In contrast, some human adapted enteric pathogens are highly infectious, with *Shigella* and EHEC colonizing people after exposure to as few as 100 CFU[22,23]. We found that the high dose required for *C. rodentium* colonization is largely attributable to the microbiota. We can contextualize this observation by considering that the microbiota of wild mice, presumably the ancestral host of *C. rodentium*, varies between individuals and differs drastically from the microbiota observed in laboratory animals[24]. Based on these observations, we speculate there could be two non-mutually exclusive reasons why *C. rodentium* was inefficient at overcoming the microbiota in our experiments: (1) this pathogen may utilize a generalist strategy, able to colonize in the context of diverse microbiotas, but not optimal to any, and/or (2) *C. rodentium* may have adapted to colonize mice whose microbiotas differ from the laboratory mice tested here. Importantly, the microbiota has been implicated in impeding enteric colonization by many human pathogens[25]. In general, heterogeneity of the microbiota within host populations may explain why the microbiota has remained a potent barrier preventing colonization by multiple pathogens.

Our results suggest that the LEE pathogenicity island is a specific adaptation strategy for *C. rodentium* (and presumably other A/E pathogens) to overcome barriers to colonization created by pressures from the microbiota. LEE activity allows attaching effacing pathogens, such as *C. rodentium*, to create a replicative niche that is in close apposition to the epithelial surface and relatively free of microbiota[3]. While the LEE is required for colonization in the presence of the microbiota, we found that this island is detrimental in the microbiota's absence. During long term colonization of germ-free mice, bacteria with genomic deletions of the region containing the LEE took over the population (Fig. 7). A similar phenomenon was previously observed in mice treated with dietary iron[26], which increased intestinal glucose and presumably decreased competition with the microbiota. These examples of selection for *C. rodentium* containing genomic deletions of the LEE suggest that the LEE has a significant fitness cost. Kamada et al. described one cost of the LEE, proposing that LEE-encoded proteins are targets of the host's antibody response, leading to the elimination of LEE expressing bacteria[27]. We propose that the LEE permits attaching and effacing pathogens to overcome microbiota-dependent pressures, but at the cost of increased stimulation of host immunity.

One way the gut microbiota is thought to inhibit intestinal colonization of invading bacteria is by competing for resources. For example, measurement of total pathogen burden has shown that *C. rodentium* colonization is limited by the availability of simple sugars and amino acids[11,12]. However, it remains unclear whether limitation of resources contributes directly to the bottleneck or impacts downstream processes such as replication. Although we did not directly address the mechanisms of the microbiota dependent bottleneck, our discovery that the number of founders increases with pathogen dose strongly suggests that the microbiota does not create a bottleneck by limiting nutrients or niches. If the microbiota creates a bottleneck by depleting an essential resource, we would anticipate an upper limit to the size of the founding population (e. g., only enough iron or binding sites to support a limited number of founders). However, we found that the size of the founding population is extremely small and increased with dose; the number of founders increased from 2 to 83 when the dose increased 100-fold from $10^8$ to $10^{10}$ CFU (Fig. 2). It seems unlikely that increasing the size of the inoculum by ~$10^{10}$ CFU would also increase the quantity of a limited resource to accommodate 81 additional founders. Thus, our findings suggest that the primary mechanism by which the microbiota prevents infection is not through limitation of a finite resource. Nonetheless, limited resources likely control other components of colonization, including pathogen expansion, carrying capacity, and/or clearance.

Our dosing studies strongly suggest that the bottleneck to *C. rodentium* infection acts by eliminating potential founders. Broadly, 'elimination' includes many mechanisms such as killing incoming bacteria or regulating their ability to establish a replicative niche. Since constriction of the founding population is slow and *C. rodentium* cells spend hours in proximity to their replicative niche without becoming founders, we favor a mechanism that impedes infectivity, rather than direct pathogen killing. We speculate that one or more soluble, microbiota-dependent metabolites are the primary factor that determines the size of the founding population. Indeed, expression of the *C. rodentium* genes required for colonization is regulated by several soluble metabolites, including bicarbonate, indole, and short-chain fatty acids[28–33].

In conclusion, we found that multiple factors contribute to the dose at which infection occurs. We propose that the mechanisms creating a bottleneck are reflected in and can largely be defined by the relationship between dose and founding population. In the case of *C. rodentium*, the most restrictive infection bottleneck is created by the microbiota, an observation that likely applies to many enteric pathogens.

## Methods

### Mice

SPF C57BL/6 J and C3H/HeOuJ mice were purchased from Jackson Laboratory. Mice were acclimated for at least 72 h prior to experimentation. Germ-free C57BL/6 J mice were provided by the Massachusetts Host-Microbiome Center and following inoculation were maintained with autoclave-sterilized cages, food, and water. Female mice 9-14 weeks of age at the start of experimentation were used. Mice were housed in a temperature (68–75 F) and humidity (50%) controlled facility with 12 h light/dark cycles. Animal studies were conducted in a biosafety level 2 (BSL2) facility at the Brigham and Women's Hospital. All experiments involving mice were performed according to protocols reviewed and approved by the Brigham and Women's Hospital Institutional Animal Care and Use Committee (protocol 2016N000416) and in compliance with the Guide for the Care and Use of Laboratory Animals.

### Bacterial Strains

A spontaneous streptomycin-resistant (SmR) mutant of *C. rodentium* strain ICC168, previously known as *Citrobacter freundii* biotype 4280 (ATCC 51459) was used to create the barcoded strains used throughout this manuscript. The growth of this SmR mutant in lysogeny broth (LB) was indistinguishable from the wild-type strain from which it was derived by growth curves and competition in liquid media. Whole genome sequencing indicates that this SmR mutant contains 2 variations from the published[34] ICC168 genome: (1) a non-coding single nucleotide variation in *tolB*, and (2) a coding single nucleotide variation in *rpsL* (which presumably confers SmR). Unless noted, *C. rodentium* was grown at 37 °C in LB broth with shaking or on solid LB agar. As necessary, media was supplemented with streptomycin (200 µg/ ml) and/or kanamycin (50 µg/ ml).

### Generation of STAMP libraries

The library "STAMP-CR253" was created by integrating a barcoded variant of the plasmid pDS132[35] into the *C. rodentium* genome in the intergenic region between *ROD_05521* and *selU*. Primers used to create STAMP-CR253 are listed in Supplemental table 2. To create the donor plasmid, we first inserted an -800 base-pair fragment of *C. rodentium* genomic intergenic sequence into plasmid pKD4 (addgene Plasmid #45605)[36] using New England Biolabs (NEB) HiFi DNA Master Mix. A fragment containing both the *C. rodentium* intergenic region and a kanamycin-resistance cassette was then amplified from the modified version of pKD4 using primers containing 20 random nucleotides (Integrated DNA Technologies). This fragment was cloned into pDS132 and a library of -2000 assembled plasmids were transformed into the donor strain MFDλpir, creating a barcoded donor plasmid library capable of integrating into the intergenic region of the *C. rodentium* genome. The donor library was conjugated to SmR *C. rodentium*, with selection for streptomycin and kanamycin resistant colonies. A tiled library of transconjugants were pooled in phosphate buffed saline plus glycerol (PBSG) and frozen at −80 °C in aliquots to create STAMP-CR253. Sequencing of the pooled library revealed the presence of 253 unique -20 nucleotide barcodes.

The library "STAMP-CR69K" was created using the plasmid donor library pSM1[7]. The pSM1 donor library is composed of -70,000 unique plasmids transformed into the donor strain MFDλpir. Each pSM1 plasmid carries a site specific Tn7 transposon containing a random -25 nucleotide barcode adjacent to a kanamycin resistance cassette. The Tn7 transposon system has previously been characterized to integrate at a neutral site in the genome downstream of the gene *glmS*[6]. Conjugation was used to introduce the pSM1 library into SmR *C. rodentium* and transconjugants containing the transposon were selected using streptomycin and kanamycin. Transconjugant colonies were pooled in PBSG and frozen at −80 °C in aliquots to create the library STAMP-CR69K. Sequencing indicated that the library contains -69,000 unique barcodes.

### STAMP sample processing

Bacteria from cultures or animal samples were plated on LB agar containing streptomycin and/or kanamycin. Colonies were resuspended in PBSG and stored at −80 °C. Genomic templates were created by diluting samples in water and boiling at 95 °C for 15 min. Barcodes were amplified from the genomic templates using either Phusion DNA polymerase or OneTaq HS Quick-Load (New England Biolabs) and amplified by PCR for 25 cycles (primer sequences are listed in Supplemental table 3). The presence of amplicons was verified by agarose gel electrophoresis. Samples were pooled, purified using the GeneJet PCR purification kit (Fisher), and sequenced with an Illumina MiSeq. FASTQ files were generated by Illumina's proprietary analysis pipeline (FASTQ Generation V1.0.0). Sequencing reads were subsequently processed using CLC Genomics Workbench (Qiagen) and Geneious (Biomatters) to determine the number of reads from each sample that map to the barcodes present in the respective libraries. The size of the founding population was determined by comparing the frequency of barcodes to the inoculum. STAMP-CR253 ($N_r$) was analyzed using the STAMP analysis pipeline[5]. To accommodate an increased number of barcodes, STAMP-CR69K ($N_s$) was analyzed using the STAMPR analysis pipeline[7,8]. Standard curves were used to refine the measurements of founding population ($N_r'$ and $N_s'$). Cavalli-Sforza chord distance was used to compare the genetic distance of barcoded samples[5].

### C. rodentium whole genome sequencing

Genomic DNA was extracted from *C. rodentium* colonies using the GeneJet genomic DNA purification kit (Thermo Fisher). Genomic sequencing was performed by the Microbial Genome Sequencing Center LLC (MiGS; Pittsburgh PA) using an Illumina NextSeq 2000. FASTQ files were generated by Illumina's proprietary analysis pipeline (FASTQ Generation V1.0.0). CLC Genomics Workbench (Qiagen) was used to map sequencing reads to the *C. rodentium* ICC168 reference genome (NCBI ASM2708v1, [https://www.ncbi.nlm.nih.gov/assembly/GCA_000027085.1] RefSeq sequence: Chr. NC_013716.1, pCROD1 NC_013717.1, pCROD2 NC_013718.1, pCROD3 NC_013719.1) and basic variant detection and structural variant detection functions were used to compare between clones. We manually sorted detected variants and only report high confident variants, defined as being present in >90% of all reads.

### Mouse infection

Unless otherwise noted, libraries of barcoded bacteria were prepared for oral challenge by resuspending frozen aliquots in liquid LB broth and expanding at 37 °C with shaking for 3–5 h. Following growth bacteria were pelleted, resuspended in phosphate buffered saline (PBS), diluted to the desired concentration, and stored at room-temperature until inoculation.

Mice were deprived of food for 3–4 h prior to inoculation. For intra-gastric gavage mice were lightly sedated with isoflurane inhalation; 100 µL of bacteria were inoculated into the stomach using a 1 mL syringe and a sterile, 18 G, 1.5 inch, 2 mm ball, feeding needle (Cadence Science). Following inoculation, dose was determined by serial dilution and plating. Mice weight, body condition, and fecal appearance were monitored during infection.

*C. rodentium* populations were isolated from fecal pellets and/or organs. Organ "Adherent" and "Luminal" populations from the cecum

and colon were separated by washing the respective organs with PBS. Samples were homogenized in PBS using a bead beater (BioSpec Products, Inc) and 2 stainless-steel 3.2 mm beads. To determine the total burden and size of the founding population *C. rodentium* were expanded from homogenized animal samples on solid media containing streptomycin and/or kanamycin.

## Loxtidine treatment

Loxtidine was generously provided by James R. Goldenring, M.D., Ph.D. (Vanderbilt University School of Medicine). Loxtidine was dissolved in PBS and 1 mg was administered intraperitoneal 3–5 h prior to inoculation with *C. rodentium*.

## Measurement of stomach acidity

The pH of stomach content was determined by aspirating liquid from the stomach immediately post-mortem and measuring the pH using a calibrated Orion Micro pH Electrode (Thermo Scientific).

## Germ-free infection

Germ-free mice were provided by the Massachusetts Host-Microbiome Center in autoclave-sterilized cages and immediately inoculated with *C. rodentium*. Mice were maintained and handled using sterile equipment.

## Streptomycin-treatment

Streptomycin-treated mice were given filter-sterilized water containing 5 mg/mL streptomycin sulfate (Teknova) for the 3 days prior to inoculation. During streptomycin treatment feces were monitored on LB agar with streptomycin for streptomycin-resistant microbes, which did not appear. Mice were switched back to regular water at the same time as they were inoculated with *C. rodentium*.

## Quantification and statistical analysis

Statistical analyses were performed using GraphPad Prism version 9. Information regarding the number of samples and statistical tests are described in the figure legends. Geometric means, geometric standard deviations, and non-parametric tests were used when quantifying bacterial populations. As the bottom of our detection limit was near 1 we substituted a value between 0.1 and 0.8 when no bacteria were detected.

## Resource availability

Requests for further information, resources, or reagents will be fulfilled by the Lead Contact, Matthew K. Waldor (mwaldor@research.bwh.harvard.edu).

## Reporting summary

Further information on research design is available in the Nature Portfolio Reporting Summary linked to this article.

## Data availability

Sequencing reads from bacterial whole genome sequencing are available in the Sequencing Read Archive under BioProject PRJNA895384. *C. rodentium* ICC168 reference genome NCBI ASM2708v1 [https://www.ncbi.nlm.nih.gov/assembly/GCA_000027085.1]. Source data are provided with this paper as a Source data file. Source data are provided with this paper.

## Code availability

STAMP scripts and barcode read counts are deposited online at [https://github.com/hullahalli/stampr_rtisan][37].

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

## Acknowledgements

We acknowledge members of the Waldor lab for their discussions and input. We thank Dr. James R. Goldenring for providing Loxtidine and Ms. Tiffany S. Davanzo (Slaybaugh Studios) for assisting with the illustrations in Figs. 1a, 2a, and 4h. Gnotobiotic mouse studies were supported by the Massachusetts Host-Microbiome Center, with funding from the NIH (P30DK034854), and the Massachusetts Life Sciences Center (MLSC). This work was supported by NIH grants to J.R.T. (R01DK61931, R01DK68271) and M.K.W. (R01AI042347), a US Department of Defense CDMRP contract to J.R.T. (PR181271), the Howard Hughes Medical Institute (HHMI) to M.K.W., and fellowships from the NIH to K.H. (F31AI156949) and I.W.C. (T32DK007477-37). This article is subject to HHMI's Open Access to Publications policy. HHMI lab heads have previously granted a nonexclusive CC BY 4.0 license to the public and a sublicensable license to HHMI in their research articles. Pursuant to those licenses, the author-accepted manuscript of this article can be made freely available under a CC BY 4.0 license immediately upon publication.

## Author contributions

I.W.C., K.H., J.R.T., and M.K.W. were involved in research design and discussion of results; I.W.C. performed experiments; I.W.C. and K.H. analyzed data; I.W.C. and M.K.W. wrote the manuscript with input and approval from all authors.

## Competing interests

The authors declare no competing interests.
