## [Peer Review File · Nature Communications]

REVIEWER COMMENTS

Reviewer #1 (Remarks to the Author):

This manuscript is well written and addresses an important question regarding host bottlenecks to establishing infection by the model enteric pathogen *Citrobacter rodentium*. Using barcoded *C. rodentium* they show that infection is due to a few unique founder cells from the inoculum. They demonstrate that there are two bottlenecks to infection that occur in the host. The first is stomach acid which reduces founder bacteria by 100-fold. However, many bacteria can pass through the stomach acid and establish replicating niches within the cecum and colon. These undergo a second bottleneck due to the gut microbiota reducing founders by 10⁵-fold after 5 days of infection. After ~20 days post-infection, the absence of the gut microbiota results in the loss of a mobile genetic element required for infection in the presence of the gut microbiota, highlighting the bottleneck effect of the gut microbiota. I recommend this article be published with a few revisions.

1. Line 140-141. I overall agree with the conclusions drawn that “the number of founders is likely not dictated by limited space or resources, contradicting the finite resource hypothesis.” However, the drop-off in founders for later time points with higher inoculum (10¹⁰ in 2B) suggests some aspect relevant to the finite resource hypothesis at least within the *C. rodentium* population. This is addressed later in lines 287-296 for the experiments in Figures 6 & 7 and the discussion in lines 377-383. However, the separation between the data and the explanation is too long and leaves the reader wondering about the discrepancy. In addition to referencing the model in 2A, pointing out an instance where this may occur in 2B in lines 287-296 would help the reader.

2. Lines 240-251. For readers unfamiliar with these mouse lines, it would be helpful to include the rationale behind using the C3H/HeOJ mice.

3. Figure 4G. Organ content does not match the labels used in the text or figure legends. All references in the text use luminal.

4. Supplemental figure 2. I recommend a qualifier for this data. It is mentioned in the text that infection requires $\geq 10^8$ CFU. It would be helpful to the reader to reference the discrepancy in what the figure legend says to what occurs with the 10⁷ dose.

Reviewer #2 (Remarks to the Author):

In this work by Campbell et. al., dubbed “Quantitative dose-response analysis untangles host bottlenecks to enteric infection” the authors investigated the relationship between pathogen inoculation dose and the size of the infection founders using barcoded populations of *C. rodentium* bacterium and its natural host, mice. The authors conclude that the size of founding population is severely constrained by host bottleneck and roughly linearly scales up with the dose. In B6 mice, inoculum of over 10^7 CFU is required to establish infection. Further, the authors determine that the scarcity of niches or resources does not appear to restrict the number of founders, since dose increase up to 10^{10} CFU led to a proportional increase in the size of founders. The authors identify host microbiota as the most restrictive factor controlling the colonization bottlenecks of the host.

This is a well-structured paper that presents important conclusions. However, in my view, two not unrelated points require clarification prior to publications.

1. Based on the in vitro dilution assay to determine the resolution limit of the barcoded libraries for the determination of a true number of founders (Figure S1B), it appears that the number of founders can be precisely determined for up to 10^4 CFU. It is probably possible to extend the limit up to 10^6 CFU by extrapolating the linear correlation. However, the inoculation experiments performed with mice were done with inoculum size exceeding the resolution limit by several orders of magnitudes ($10^7 - 10^{10}$ CFU). If so, how the determination of the founder size was achieved?

2. Based on the linear correlation maintained between the dose and the number of founders (when the inoculum size was increased up to 10^{10} CFU), the authors conclude that niche scarcity and limited resources do not play a significant role in colonization bottlenecks. However, in my view, this conclusion is bluntly contradicted by the finding that microbiota severely restricts the size of the founder population. Isn't competition with microbiota plainly means competition for niches and resources? If so, can it be that the proportional decrease in the founder size was not observed with the increase in dose simply because the founder size was overestimated (see point 1 above)?

Reviewer #3 (Remarks to the Author):

In this paper, the authors used infectious disease studies in mice, along with bacterial population analysis to delineate where, by what mechanisms, and to what extent bottlenecks occur during the colonization of mice with pathogenic bacteria, by using the model enteric pathogen *Citrobacter rodentium*. Notably, the authors infected mice with 10-fold increments of a known population of a barcoded *Citrobacter rodentium* library (STAMP) and compared the input population with the

recovered bacteria both temporally and longitudinally. The results support the hypothesis that the bottleneck was due to the mechanisms of elimination, rather than due to finite resources since the increased doses resulted in a proportionally increased founding population. Importantly, the authors provide in vitro and in vivo data that indicates that pH of the stomach contributes between 10 to 100-fold to the bottleneck. Additionally, the authors demonstrate that fecal samples contain highly similar bacterial populations to those found in the cecum and colon. Finally, the author's key results show that after streptomycin treatment (depletion of microbiota), and in germ-free mice, the bottleneck is strikingly minimized, provide strong evidence that the microbiota imparts a severe bottleneck to the bacteria that manage to survive passage within the stomach. Overall, the conclusions and significance are original, and the results are of immediate interest to a broad range of research fields involving enteric pathogenic bacteria in the Enterobacteriaceae, as well as various other enteric pathogens. The approach and quality of the data and presentation is appropriate and applies statistical tests appropriately.

Major Comments:

Lines 253-322. Streptomycin treated and germ-free mice show a minimal bottleneck compared to conventional mice, indicating that the microbiota contributes significantly to the bottleneck. However, the authors do not show if the pH of germ-free mouse stomachs is equally or less acidic than conventional mice, due to losing microbiota, or not having microbiota to begin with. Lacking microbiota likely influences food intake, and theoretically, stomach pH. Demonstrating that the pH of Streptomycin treated, germ-free, and conventional mice does not vary significantly upon infection would strengthen this hypothesis significantly.

Minor Comments:

Line 206: The section on super colonization resistance does not fully articulate its significance or contribution to the focus of the paper, and instead seems like the subject for a separate paper involving the mechanisms of super-colonization resistance. Nevertheless, this section may indeed strengthen the concept that the primary bottleneck occurs at sites of infection, but if so, the section would benefit from more direct wording to highlight this concept.

Reply to review comments:

(review in black and reply in blue)

We thank the reviewers for their time and efforts considering our paper.

Reviewer #1 (Remarks to the Author):

This manuscript is well written and addresses an important question regarding host bottlenecks to establishing infection by the model enteric pathogen *Citrobacter rodentium*. Using barcoded *C. rodentium* they show that infection is due to a few unique founder cells from the inoculum. They demonstrate that there are two bottlenecks to infection that occur in the host. The first is stomach acid which reduces founder bacteria by 100-fold. However, many bacteria can pass through the stomach acid and establish replicating niches within the cecum and colon. These undergo a second bottleneck due to the gut microbiota reducing founders by 10⁵-fold after 5 days of infection. After ~20 days post-infection, the absence of the gut microbiota results in the loss of a mobile genetic element required for infection in the presence of the gut microbiota, highlighting the bottleneck effect of the gut microbiota. I recommend this article be published with a few revisions.

We thank the reviewer for their careful consideration of our data and constructive suggestions.

1. Line 140-141. I overall agree with the conclusions drawn that “the number of founders is likely not dictated by limited space or resources, contradicting the finite resource hypothesis.” However, the drop-off in founders for later time points with higher inoculum (10¹⁰ in 2B) suggests some aspect relevant to the finite resource hypothesis at least within the *C. rodentium* population. This is addressed later in lines 287-296 for the experiments in Figures 6 & 7 and the discussion in lines 377-383. However, the separation between the data and the explanation is too long and leaves the reader wondering about the discrepancy. In addition to referencing the model in 2A, pointing out an instance where this may occur in 2B in lines 287-296 would help the reader.

Thank you for the close attention to our data. In Figure 2B, we observe that the founding population of the mice inoculated with the highest dose (10¹⁰ CFU; blue) decreases from 39 founders on day 11 to 2 founders on day 16 (geometric means). This drop-off in the size of the founding population occurs during clearance of the pathogen, when there is a decrease in the total size of the population (burden). The most likely explanation for this simultaneous contraction of both the total population and founding population is antibody related elimination of the pathogen (as shown in Maaser, Housley, ..., and Eckman, 2004). This phenomenon is distinct from the observation in Figures 6 and 7, where, in the absence of a microbiota the founding population decreases over a period of 20 days without a decrease in the total size of the population, suggesting a combination of pathogen evolution and intra-pathogen competition. Importantly, neither of these observations are consistent with finite resources creating the initial bottleneck to colonization, which occurs over the first 24-48 hours.

We have added additional language to the description of Figure 2 to clarify this point.

2. Lines 240-251. For readers unfamiliar with these mouse lines, it would be helpful to include the rationale behind using the C3H/HeO_uJ mice.

Thank you for this suggestion. We added additional language to this section to clarify that our rationale was to quantify the bottleneck in a more disease susceptible strain to test the magnitude of the host's contribution to the infection bottleneck.

3. Figure 4G. Organ content does not match the labels used in the text or figure legends. All references in the text use luminal.

Thank you for alerting us to this discrepancy. We have fixed the labels in the display item such that every reference to these data is now described as "luminal".

4. Supplemental figure 2. I recommend a qualifier for this data. It is mentioned in the text that infection requires $\geq 10^8$ CFU. It would be helpful to the reader to reference the discrepancy in what the figure legend says to what occurs with the 10^7 dose.

Thank you for helping us clarify this figure. We have added a qualifier to the Supplemental figure 2 legend that should assist the reader in understanding the 10^7 CFU dose.

Reviewer #2 (Remarks to the Author):

In this work by Campbell et. al., dubbed "Quantitative dose-response analysis untangles host bottlenecks to enteric infection" the authors investigated the relationship between pathogen inoculation dose and the size of the infection founders using barcoded populations of *C. rodentium* bacterium and its natural host, mice. The authors conclude that the size of founding population is severely constrained by host bottleneck and roughly linearly scales up with the dose. In B6 mice, inoculum of over 10^7 CFU is required to establish infection. Further, the authors determine that the scarcity of niches or resources does not appear to restrict the number of founders, since dose increase up to 10^{10} CFU led to a proportional increase in the size of founders. The authors identify host microbiota as the most restrictive factor controlling the colonization bottlenecks of the host.

This is a well-structured paper that presents important conclusions. However, in my view, two not unrelated points require clarification prior to publications.

We thank the reviewer for their positive assessment and for helping us identify these points for clarification.

1. Based on the in vitro dilution assay to determine the resolution limit of the barcoded libraries for the determination of a true number of founders (Figure S1B), it appears that the number of founders can be precisely determined for up to 10^4 CFU. It is probably possible to extend the limit up to 10^6 CFU by extrapolating the linear correlation. However, the inoculation experiments performed with mice were done with inoculum size exceeding the resolution limit by several orders of magnitudes ($10^7 - 10^{10}$ CFU). If so, how the determination of the founder size was achieved?

Thank you for alerting us to this point of confusion. It is critical to note that in STAMP the size of the inoculum (dose) does not impact the quantification of the founding population size. Dose and founders are measured by separate techniques. The total size of a bacterial population, such as dose or burden, is measured by serial dilution and plating to determine the number of colony forming units (CFU). Dose does not have a detection limit and can be accurately measured to values exceeding 10^{10} CFU. Founders are measured by sequencing the barcodes present within a sample to determine the number of founders (N_r or N_s) that gave rise to the observed population. Founding population has a resolution limit of $\sim 10^6$ N_r or N_s , as measured in Supplemental figure 1.

To help readers we have included text in the results, figure legends, and methods indicating the methodological distinction between the measurement of dose and founding population. Dose, founders, and burden often appear together in display figures as it is useful to track the initial (dose), minimum (founders), and final (burden) size of the population on the same graph. To avoid confusion, we have labeled dose with "dose (CFU)" and the founders with "founders (N_r or N_s)" to distinguish them as measurements arising from separate techniques.

2. Based on the linear correlation maintained between the dose and the number of founders (when the inoculum size was increased up to 10^{10} CFU), the authors conclude that niche scarcity and limited resources do not play a significant role in colonization bottlenecks. However, in my view, this conclusion is bluntly contradicted by the finding that microbiota severely restricts the size of the founder population. Isn't competition with microbiota plainly means competition for niches and resources? If so, can it be that the proportional decrease in the founder size was not observed with the increase in dose simply because the founder size was overestimated (see point 1 above)?

Thank you for this interesting discussion point.

We do not dispute that competition with the microbiota for scarce resources plays a role in *C. rodentium* population dynamics during infection. As the reviewer correctly points out the microbiota is known to control pathogen expansion and clearance through competition for resources (Kamada, Kim, ..., and Nunez, 2012). However, our findings conclusively show that scarcity of resources does not impact the earliest step in infection – determining the size of the founding population. As discussed in point 1 above, we are highly confident in our calculations of founding population and our conclusion that increasing dose increases the size of the founding population. In particular, in Figure 2C it is evident that we are not reaching our resolution limit for detecting founding population at 5 days post inoculation because we can quantify more numerous founding populations at earlier timepoints from the same experiment (Figure 2B). Given the data in Figure 2C, we conclude that the size of the founding population increases with dose and that the number of founders is thus unlikely to be controlled by the scarcity of limited resources or niches; i.e., if the size of the founding population was determined by the scarcity of resources/niches, founding population would be fixed at the limit dictated by the scarce resource and therefore would not increase with dose. Given these findings, we propose that a microbiota dependent factor likely creates the infection bottleneck by eliminating the pathogen (by killing and/or inhibiting the pathogens infectivity).

Reviewer #3 (Remarks to the Author):

In this paper, the authors used infectious disease studies in mice, along with bacterial population analysis to delineate where, by what mechanisms, and to what extent bottlenecks occur during the colonization of mice with pathogenic bacteria, by using the model enteric pathogen *Citrobacter rodentium*. Notably, the authors infected mice with 10-fold increments of a known population of a barcoded *Citrobacter rodentium* library (STAMP) and compared the input population with the recovered bacteria both temporally and longitudinally. The results support the hypothesis that the bottleneck was due to the mechanisms of elimination, rather than due to finite resources since the increased doses resulted in a proportionally increased founding population. Importantly, the authors provide *in vitro* and *in vivo* data that indicates that pH of the stomach contributes between 10 to 100-fold to the bottleneck. Additionally, the authors demonstrate that fecal samples contain highly similar bacterial populations to those found in the cecum and colon. Finally, the author's key results show that after streptomycin treatment (depletion of microbiota), and in germ-free mice, the bottleneck is strikingly minimized, provide strong evidence that the microbiota imparts a severe bottleneck to the bacteria that manage to survive passage within the stomach. Overall, the conclusions and significance are original, and the results are of immediate interest to a broad range of research fields involving enteric pathogenic bacteria in the Enterobacteriaceae, as well as various other enteric pathogens. The approach and quality of the data and presentation is appropriate and applies statistical tests appropriately.

We thank the reviewer for their positive assessment of the quality and impact of this work.

Major Comments:

Lines 253-322. Streptomycin treated and germ-free mice show a minimal bottleneck compared to conventional mice, indicating that the microbiota contributes significantly to the bottleneck. However, the authors do not show if the pH of germ-free mouse stomachs is equally or less acidic than conventional mice, due to losing microbiota, or not having microbiota to begin with. Lacking microbiota likely influences food intake, and theoretically, stomach pH. Demonstrating that the pH of Streptomycin treated, germ-free, and conventional mice does not vary significantly upon infection would strengthen this hypothesis significantly.

Thank you for suggesting this experiment. We have added data on the stomach acidity of germ free and streptomycin treated mice as Supplemental figure 5. Streptomycin treatment does not change the acidity of the animal's stomach, strengthening our conclusion. The stomach of germ-free animals trends towards a higher pH, but not drastically enough to affect our conclusions, especially given that streptomycin does not impact stomach acidity.

Minor Comments:

Line 206: The section on super colonization resistance does not fully articulate its significance or contribution to the focus of the paper, and instead seems like the subject for a separate paper involving the mechanisms of super-colonization resistance. Nevertheless, this section may indeed strengthen the concept that the primary bottleneck occurs at sites of infection, but if so, the section would benefit from more direct wording to highlight this concept.

Thank you for helping us identify this area for further clarification. We agree that this experiment points to interesting biology worthy of future inquiry. We included this experiment here because it is an important control given that all our experiments were performed with cohoused animals. This paper deeply explores the origins of the pathogen's founding population and it is therefore important to exclude the possibility that the founding population originates from other cohoused mice, rather than the initial inoculum. We have added additional text to this section to help clarify the data's contribution to the rest of the paper.